# Evidence-Based Application of Acupuncture for Pain Management in Companion Animal Medicine

**DOI:** 10.3390/vetsci9060252

**Published:** 2022-05-26

**Authors:** Janice L. Huntingford, Michael C. Petty

**Affiliations:** 1Essex Animal Hospital, 355 Talbot St N, Essex, ON N8M 2W3, Canada; 2Chi University, 9650 W Hwy 318, Reddick, FL 32686, USA; 3Arbor Pointe Veterinary Hospital, Animal Pain Center, 42043 Ford Rd, Canton, MI 48187, USA; mike.c.petty@gmail.com

**Keywords:** acupuncture, electroacupuncture, chronic pain, TCVM, mechanotransduction, neurophysiology

## Abstract

The use of veterinary acupuncture for pain relief is expanding among small animal practitioners. Although acupuncture was developed as part of the medical system in Ancient China, research into the scientific basis of its effects is expanding rapidly. Acupuncture is very effective for analgesia on a local, segmental, and suprasegmental level. Many forms of acupuncture can be used independently or as part of a balanced multi-modal approach for the control of acute and chronic pain. In the hands of a skilled practitioner, acupuncture can be a safe and effective modality for treating pain in companion animals. This article outlines the mechanisms of action of acupuncture, its related neurophysiology and provides examples from the literature demonstrating its effectiveness.

## 1. Introduction

Traditional Chinese Medicine (TCM) has been practiced for more than 2000 years. It is based on the philosophy that wellness exists in balance and disease is caused by an imbalance of energies within the body. Acupuncture is one of branches of TCM and Traditional Chinese Veterinary Medicine (TCVM). It is a highly effective method to treat pain. [1,2]. The use of veterinary acupuncture for pain relief is expanding among small animal practitioners. It is used as an adjunct to conventional practices, as an alternative when conventional treatments fail to resolve the patient’s pain or when a client is seeking a drug-free way to mitigate pain [1,3,4,5]. The most recent canine and feline pain management guidelines, developed by the World Small Animal Veterinary Association Global Pain Council include acupuncture as a non-pharmaceutical treatment for small animal pain (https://wsava.org/global-guidelines/global-pain-council-guidelines, accessed on 21 November 2021).

Research into the scientific basis of acupuncture is expanding as evidenced by the growing number of scientific papers published on this topic. Of these roughly 4000 papers, 41% are focused on acupuncture for analgesia or pain relief [6] and the majority concerned humans or laboratory animals [7,8,9,10,11]. Nevertheless, these principles can be integrated into clinical veterinary practice. There have been some recent clinical studies in companion animals showing the effectiveness of acupuncture at mitigating pain [12,13,14,15,16,17,18].

Despite the increase in research and pet owner acceptance, some veterinary practitioners are reticent to integrate acupuncture into their practices because they believe it is based on an antiquated system [19]. TCVM acupuncture does use metaphorical language to describe patterns of disease, treatment, and pathophysiology; however, the terminology can be translated into physiological and biomedical principles that can be understood by the conventionally trained veterinarian. Acupuncture has evolved into a medically appropriate, scientifically driven treatment for pain in companion animals [3].

## 2. Acupuncture Points and Meridians

Acupuncture involves the stimulation of specific anatomical locations called acupuncture points or acupoints to create a local, segmental and general physiological effect that is mediated by neuromodulation [19,20]. Acupoints are neurovascular bundles consisting of a concentrated area of free nerve endings, lymphatics, small arterioles, venioles and mast cells. Areas that are richly innervated and areas with high numbers of somatic afferent fibers frequently contain many acupuncture points [19]. Acupuncture points are divided into Four Types (See Table 1) and have a higher electrical conductance, lower impedance and a higher capacitance than the surrounding tissue. Many of these points are located in palpable depressions and in areas where pain and muscle dysfunctions generate myofascial trigger points [21].

Acupuncture points may be stimulated by the insertion of a tiny filiform needle, application of heat (moxibustion) or pressure (acupressure), use of laser light, friction and cupping. In addition, practitioners may elect to inject the acupoints with substances to induce a longer stimulation of the points. Examples of this are injections of vitamin B 12 or implantation of gold beads into acupuncture points [1,2]. Acupuncture needles may be stimulated manually by twisting or moving the needle during the treatment or may be stimulated by the use of electroacupuncture (EA). EA involves attaching an electrode to a pair of acupuncture points and running a small electric current through them. It is thought to provide a stronger and longer-lasting stimulation [1,2,21].

When a needle stimulates an acupuncture point, it activates neural and neuroactive components. These activated components that are found within the skin, connective tissue and muscle surrounding the inserted needed have been defined as neural acupuncture units (NAUs) [22]. With the variety of acupuncture points, it is important to balance any acupuncture treatment with local and remote points to address different sources of pain.

In TCVM, acupuncture points represent an area of concentrated Qi (energy or life force). The stimulation of these points encourages a smooth and harmonious flow of Qi along the meridians or channels, which link the acupuncture points throughout the body. There are 12 regular or principal meridians, which are named after Chinese or Zhang Fu organs. The Zhang Fu organs have Western names but are not considered true anatomical structures. Rather, they are a collection of organs that produce and regulate Qi. These organs have interconnected functions that explain how Qi is produced and flows through the body [23]. The blockage of Qi or nerve energy causes pain. The manipulation of one acupuncture point may cause an effect in an area remote to the site but further down the meridian. The Meridians have names such as Bladder and Gall Bladder and the acupuncture points assigned numbers along the meridian. In addition to the twelve regular Meridians, there are eight Extra Meridians, and these have specific functions. The eight Extra Meridians maintain the homeostatic balance in the body by storing, regulating, and receiving excess pathogenic Qi, balancing the metabolism and supporting blood, fluids and Qi [1,2].

Extra meridian points are found on these Meridians and often have classical Chinese names such as *Shen Shu*. Both TCVM acupuncture and Western medical acupuncture use the same names for points to designate where the needles are placed [23]. See Figure 1 and Figure 2 for examples.

Whereas TCVM acupuncture looks at restoring the balance of Qi, Western Medical acupuncture looks at restoring physiologic homeostasis and utilizing mechanisms within an animal to relieve pain [24]. From this perspective, meridians are thought to be consistent with the peripheral nervous system, run along fascial planes, involve the lymphatics and may involve the interstitial space [25]. Physiologically, all acupuncture points share the same effects including fibroblast mechanotransduction, fascial interaction, neuromodulation leading to analgesia, fluid motion modulation and the manipulation of the microvascular environment. [23,26,27,28]. Some points may specifically cause immune modulation, affect the autonomic nervous system, visceral organs, or the brain, modulate lymphatic flow or cause neuromuscular effects [19].

**Figure 2 vetsci-09-00252-f002:**
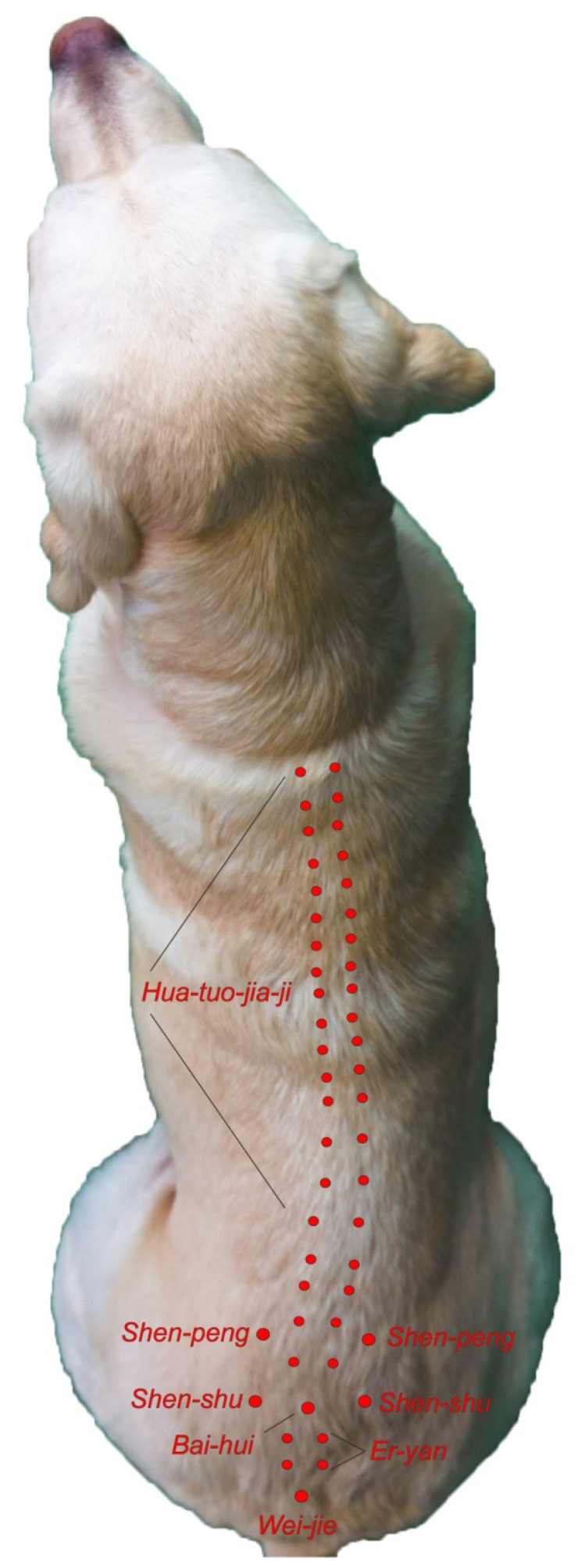
Extra Meridian Points with Classical Chinese Names. Reprinted with permission from Ref. [29]. 2018, Dewey C and Xie S.

## 3. Analgesic Effects of Acupuncture

### 3.1. The Pain Pathway

To better understand how acupuncture is effective in treating pain, it is helpful to review the physiology of pain. The pain pathway, which includes transduction, transmission, modulation and perception is part of the physiology of pain. The first step in the pain pathway occurs when damaged cells release inflammatory mediators such as interleukins (IL), prostaglandins (PG), arachidonic acid (AA), serotonin, substance P, bradykinin, and glutamate. Transduction occurs when nociceptors convert this chemical signal into an electrical impulse [30].

Transmission involves sending an electrical impulse to the brain via the afferent nerve fibers. The fibers involved are the A-beta, A-delta and C fibers. See Table 2 for a comparison of the different types of nerve fibers. A-beta fibers are responsible for low-threshold or non-painful information, whereas A-delta fibers are responsible for the withdrawal reflex and sharp acute pain. C fibers, which transmit their information at a slower rate, are responsible for slow, burning pain which can intensify. The stimulation of C fibers has been implicated in the development of chronic pain [30].

Modulations in the nerve signal occur in the dorsal horn (DH) of the spinal cord. The impulses from the A-Delta and C fibers change in the DH and are either inhibited or amplified. Excitatory and inhibitory neurotransmitters transform the signal before it reaches the brain. Transmitters such as glutamate intensify the pain signal whereas gamma-amino butyric acid (GABA) acts as an inhibitor. Serotonin can suppress the action of glutamate while enhancing that of GABA. Endogenous opioids are also released and influence the pain signal. Opioid receptors are triggered to suppress excitatory neurotransmission. A-beta nerve fibers also act as a gate control to suppress pain signals in the DH [30].

Perception is the final step in the pain pathway, and it involves the recognition and integration of pain. Multiple brain regions and ascending pathways are involved in the processing of pain. The perception centers in the brain control the overall autonomic, emotional and physical response to pain [30].

Acupuncture analgesia can be divided into local, segmental (spinal) or suprasegmental effects [21]. Local effects are mediated through the sensory nociceptors stimulating the afferent nerves whose cell bodies are in the dorsal root ganglia. Spinal effects are mediated by neurons in the dorsal gray matter of the spinal cord. These are second-order neurons which provide the ascending nociceptive tracts to the brain. In the brain, a descending pain-mitigating system (suprasegmental) can be stimulated to suppress the pain signal sent by the second-order neurons, thus providing analgesia [21].

### 3.2. Local Effects of Acupuncture

Local analgesic effects start immediately at the site of needle insertion and act as a form of counter-irritation. Collagen and elastin fibers wind around the needle and this acts on fibroblasts as well as local nerve endings and vasculature. The physical sensation of the needle causes the fibroblasts to release biochemical substances that elicit a response (mechanotransduction) [27,28]. Data suggest that the deeper the needle is inserted, the more dramatic the effects [31]. Neuronal signalling and needle insertion are intimately connected. When the fascia and tissue matrix is deformed by the needle, the local axon reflex is triggered in the nerve endings. This response is thought to cause the De Qi sensation which humans describe as dull or achy and travels from the site of initial needle insertion to the nerves. The De Qi response is associated with the effectiveness of the treatment and leads to the release of neuropeptides such as calcitonin gene-related peptide, nerve growth factor, substance P and many others [19,32].

The needle creates microtrauma or local tissue damage which activates Hageman’s tissue factor XII. This in turn results in the activation of the local coagulation cascade and the complement cascade, leading to the production of plasminogen, protein kinins, and prostaglandins. The microtrauma also causes mast cell degranulation, which releases histamine, heparin, proteases, and bradykinin. These local reactions ultimately result in increased blood flow to the area and local immune responsiveness, which help to relieve pain and reduce inflammation and edema. Connective tissues become stretched when needles are placed in acupoints, which in turn relaxes the muscles and tissues in the local area [19,32].

Many of the local effects of acupuncture are thought to be mediated by endogenous opioids. Tissue lymphocytes, macrophages and granulocytes release endogenous opioids when stimulated by an acupuncture needle. The peripheral sympathetic nerve receptors are acted on by these opioids to suppress the tissue nociceptors and to release more endogenous opioids into the area [19]. Studies in both humans and laboratory animals have shown increased levels of endogenous opioids in the plasma and CSF after acupuncture treatments [32]. A pilot study in dogs undergoing ovariohysterectomies compared dogs that received electroacupuncture for pain control versus those who received butorphanol. Both treatments were effective for pain control and the acupuncture group did not require any rescue pain control, although the butorphanol group did [33].

Other effects observed as a result of acupoint stimulation include an increase in cannabinoid CB2 receptors, which causes the upregulation of endogenous opioids, inhibition of cyclooxygenase-2 and prostaglandin production, and upregulation of TRPV-1 receptors in peripheral nerve endings [19]. A cascade of local and systemic effects ultimately provides analgesia to the patient for painful conditions. Painful local clinical conditions such as burns, lick granulomas, chronic wounds and local soft tissue injuries can benefit from the local analgesic effects of acupuncture [19].

### 3.3. Spinal or Segmental Effects of Acupuncture

Acupuncture-based modulation of pain is complex and involves many ascending and descending pain pathways, receptors, and molecules [19,32]. The gate control theory proposed by Melzack and Wall in 1965 is the main theory relating to the segmental mechanism of pain relief and explains pain modulation in the spinal cord [34]. The pain gate is located in the dorsal horn of the spinal cord. The substansia gelatinosa (SG) cells found in the dorsal horn control the transmission of pain impulses from the periphery to the CNS. Non-noxious stimulus can occur by needling or manual pressure, which stimulates the A-beta fibers and effectively closes the SG pain gates to painful input and prevent the pain signal from travelling further up to the CNS [34]. The main structures in the spinal cord that are involved in segmental pain relief are the A-Delta and C-ascending pain fibers and the A-beta fibers which effectively close the pain gate. See Table 1. Gate control theory says that a high volume of input from the A-Beta fibers effectively closes the gate in the SG to the pain impulses carried by the C and A-Delta fibers [19].

In addition to the gating of the nerve fibers in the spinal cord, other mechanisms and molecules are involved in segmental pain relief. N-methyl-d aspartate (NMDA) is a receptor for the excitatory neurotransmitter glutamate which is released with the stimulation of nociceptors. The activation of the NMDA receptors in the DH has been associated with hyperalgesia and neuropathic pain. Acupuncture has been shown to decrease the NMDA receptor activity in the dorsal horn neurons by increasing the levels of serotonin, noradrenaline, and endogenous opioids [19,32,35,36,37]. Increased serotonin levels in the spinal cord enhance the ability of gamma aminobutyric acid (GABA) to suppress the transmission of the pain signal [19,35]. Acupuncture stimulation reduces spinal glial cell activation and suppresses the release of inflammatory cytokines such as Il-1 B, IL-6, TNF a, COX-2 and PGE2 [19]. Substance P, which activates glial cells, is also suppressed by acupuncture while GABA release is facilitated by acupuncture. Increased GABA in the extracellular space is associated with anti-nociceptive activity. Just as in the periphery, acupuncture decreases the expression of COX-2 in the spinal cord, thereby reducing wind up pain [19,35].

Spinal segmental dysfunction is a term used by medical acupuncturists, physiatrists, physiotherapists, chiropractors and osteopaths. It has been defined as a functional problem in the somatic system, which may include skeletal, arthrodial, or myofascial structures and their related vessels, lymphatics or nerves [38]. The spinal cord segments and vertebral column are closely related with the roots and spinal nerves distributed throughout the vertebral column. Dermatomes exist for each spinal nerve and studies have shown that there is a considerable overlap of up to three dorsal roots in the lumbosacral area as well as an overlap of the cutaneous area of peripheral nerves. When treating these segments with acupuncture, any part of one segment will treat the entire segment [32]. Segmental dysfunction is a significant cause of pain in small animals and affects all components of the musculoskeletal system. It is not a true disease but often manifests as weakness, asymmetry, tenderness, decreased range of motion in joints, myofascial pain, and pathologic muscle tone. There is some evidence that segmental dysfunction is mediated by the sympathetic nervous system [32].

The segmental effects of acupuncture extend to visceral analgesia as well as musculoskeletal and neurological analgesia. Visceral and somatic afferent fibers converge in the dorsal horn. According to de Lahunta, visceral referred pain is “referred to the surface of the body innervated by the general somatic afferent (GSA) neurons whose axons terminate in the same spinal cord segment and on the same neuronal cell bodies as the general visceral afferent neurons” [39]. If the appropriate somatic receptors at the segmental level are stimulated, then visceral pain can be suppressed.

Acupuncture has been shown to alter blood flow and autonomic activity that regulates visceral pain and function. Functional gastrointestinal disorders such as gastrointestinal motility and stomach acid secretion can be altered by acupuncture via somatosympathetic or somatoparasympathetic pathways depending on the acupoints stimulated [32].

### 3.4. Suprasegmental (Brain) Effects of Acupuncture

The effects of acupuncture on pain perception in the brain of companion animals are not as well understood as the segmental effects. Endogenous pain management primarily occurs through the stimulation of an anti-nociceptive network that projects to the dorsal horn from several different brain segments [19,31]. According to Dewey and others, the brain segments involved include the cerebral cortex, the hypothalamus, the periaqueductal gray (PAG) matter, the pons and parts of the medulla [19,32].

Chemical messengers involved in the suppression of pain perception include Beta endorphins, enkephalin, noradrenaline, dopamine and serotonin. Part of this network also includes the pituitary gland, which releases oxytocin and adrenocorticotrophic hormones, both of which have analgesic effects [20]. Human magnetic resonance imaging (MRI) studies have shown that when specific acupuncture points are stimulated, specific regions of the brain are activated or deactivated. Functional magnetic resonance imaging (fMRI) measures brain activity by detecting changes in blood oxygenation and flow that occur in response to neural activity. Several fMRI studies have shown that the limbic system, hypothalamus, and arcuate nucleus are all stimulated by acupuncture [40,41,42]. When the hypothalamus is stimulated by acupuncture, it releases Beta endorphins and in turn activates the PAG which has been shown to be the primary CNS structure responsible for descending pain inhibition [19,32]. Needling limb points has been shown to activate the intended brainstem center without producing a segmental effect, whereas needling a dermatomal segment such as a back Shu point for thoracolumbar pain produces a mainly segmental response or a segmental and suprasegmental response (descending brainstem response) [19]. A combination of different points can recruit local, segmental, and suprasegmental responses from the nervous system for optimum results for pain control.

Chronic stimulation can cause the CNS to alter its chemical and structural organization. This is called neuroplasticity. Maladaptive neuroplasticity can occur as a result of chronic pain that induces the CNS to facilitate pain transmission. However, acupuncture induces beneficial neuroplasticity that prevents or reverses the changes caused by chronic neuropathic pain. This neuroprotective effect is achieved via neurotransmitters, and by decreasing microglial activation and inflammation [19]. Analgesic effects produced by EA tend to be more potent and long lasting than those produced by DN alone. Low frequency EA (2–10 Hz) causes the release of endorphins and enkephalins, which are most effective for mitigating pain. Other pain relievers such as dynorphin and serotonin are released at higher frequencies and may impart an additive effect [16,19,43,44].

## 4. Clinical Applications of Acupuncture for Pain Mitigation

Acupuncture can be very effective in treating the chronic pain and decreased mobility of degenerative joint disease. The segmental effects of acupuncture can provide pain relief, improve circulation and healing, and increase mobilization [19]. Pain from stifle disease is modulated through the nerves at the L4-L6 spinal segments. When this area, as well as the muscles around the stifle are treated with dry needle acupuncture (DN), the stifle experiences significant pain relief [32]. This effect can be enhanced by the addition of electroacupuncture (EA). A retrospective study of senior dogs with cranial cruciate ligament (CCL) ruptures showed that those treated with acupuncture and Chinese Medicine recovered their stifle function within 6 to 10 months of initiating treatment, even though no surgery was performed [45]. In another study of 40 dogs with CCL disease, half of the dogs were assigned to receive acupuncture and TCVM therapy whereas the other half received conservative therapy with pain management, joint supplements and exercise therapy. Both groups of dogs showed improvements by week 24 but those treated with acupuncture attained quicker results [46].

Hip osteoarthritis (OA) is a common painful and chronic disease in medium to large breed dogs that can result from age related changes, or secondary to a congenital condition such as hip dysplasia. It is associated with significant pain and mobility issues and affects the quality of life of patients who suffer from this condition. Many clinical trials have indicated acupuncture to be useful for analgesia for these patients [47,48,49,50]. In addition to DN and EA, aquapuncture (acupoint injection) or the implantation of gold beads or wires was commonly performed [37]. In a Brazilian clinical study, nine dogs with confirmed hip dysplasia that had a poor response to medical therapy were injected either with the autologous stromal vascular fraction (SVF, *n* = 4) or allogeneic cultured adipose-derived stem cells (ASCs, *n* = 5) into three acupuncture points (BL-54, GB-29 and GB-30) near the affected joint. After 1 week, both groups demonstrated a marked improvement in lameness and pain scores which continued to improve up to day 30. Only one dog in the ASC group did not improve. This study showed that the acupoint injection of stems cells can be an effective therapy for the pain of hip OA [47].

Several studies and clinical case reports were conducted on gold bead implantation into acupuncture points around the hip for the treatment of canine hip dysplasia [48,49,50,51]. Some of these studies [48,49,50] showed positive results, although a German review showed that some of the research may have been flawed [51]. Objective outcome measures were used in one study; however, this study did not show any significant difference between the placebo group and the group treated with gold bead implantation [37].

Another interesting study compared EA and laser acupuncture (LA) for the treatment of coxofemoral OA. A total of 31 dogs were randomly assigned to two groups. The Canine Brief Pain Inventory (CBPI) was completed by the owner and the goniometric range of motion (ROM) of both coxofemoral joints was determined as the baseline for all dogs. One group received laser acupuncture with a class IV laser to hip points (BL-54, GB-29 and GB-30). A total of 12 joules (J) was delivered at each AP point. The other group received electroacupuncture to the same group of points. Each pet was treated once a week and only the owner was blinded to which therapy their pet was receiving. At the end of 8 weeks, the CBPI score was repeated by the owner while the researcher repeated the ROM measurements. Both groups showed an improvement in pain scores with the EA group having improved statistically more, whereas only the LA group showed an improvement in hip extension. Laser acupuncture appears to be a valid pain therapy for those dogs who are needle phobic [16].

Acupuncture, particularly EA, has been shown to improve analgesia and accelerate the recovery of motor functions in dogs with intervertebral disc disease (IVDD) [43]. Hayashi and colleagues conducted a study on 50 dogs with thoracolumbar IVDD of various grades. The dogs were randomised to receive either conventional medical treatment or conventional medical treatment with EA. The overall success rate, as determined by their ability to walk without assistance, was significantly higher with EA and the time to ambulation was also significantly shorter in the EA group [44]. Another study of dogs with thoracolumbar disc disease showed that when electroacupuncture was combined with Western treatment, the medium time to ambulation was shorter in those dogs that received acupuncture than in those dogs who did not [52]. A similar result was reported by Han and colleagues for a group of 80 paraplegic dogs in which half of the dogs were treated with prednisone and the remainder were treated with prednisone combined with acupuncture and electroacupuncture. The dogs who were treated with a combination of DN, EA and prednisone had significantly less pain, recovered to ambulation at a faster rate and had decreased relapses [53]. Appendix A lists common acupuncture pain points that may be of interest to the reader.

## 5. Acupuncture Contraindications and Precautions

Acupuncture is a very safe modality. Adverse events can generally be avoided through careful patient selection. Placing needles in inflamed or infected skin should be avoided. Needles should not be used in patients with bleeding disorders. In pregnant animals, points that may cause premature labor should be avoided. EA should not be applied across tumor sites, across the chest in animals with pacemakers or across the skull in animals with seizure conditions. In areas of fracture or acute trauma, needle placement should be avoided. Although a broken acupuncture needle is rare, it is possible that a migrating, broken needle could cause harm [2].

## 6. Conclusions

Acupuncture has profound effects on both the peripheral and central nervous systems. There is plenty of research concerning laboratory animals and humans, but the field of veterinary acupuncture would benefit from more validated clinical studies in companion animals. For many veterinary patients, pain management can be challenging due to medical co-morbidities and the sensitivity of many patients to pharmaceutical pain control. Acupuncture fits well into a multi-modal model for the management of acute and chronic pain and is a safe complementary modality that can be embraced by trained practitioners in companion animal practices.

## Figures and Tables

**Figure 1 vetsci-09-00252-f001:**
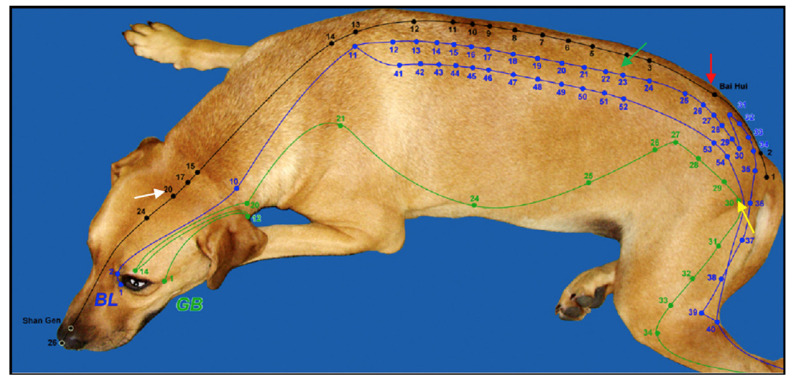
Meridians with Acupoints. Green is Gall Bladder (GB) Meridian, Blue is Bladder (BL) Meridian and Black is Governing Vessel (GV). GV-20 = White Arrow, Bl-23 = Green Arrow, Bai-Hui = Red Arrow, GB-30 = Yellow arrow. (Reprinted with permission from Ref [2]. 2003, H. Xie).

**Table 1 vetsci-09-00252-t001:** Types of Acupuncture points (Data from Reference [21]).

Type of Point	Location of Point
Type I	Motor points Located in areas where nerves enter muscles 67% of all acupoints are motor points
Type II	Located on superficial nerves in the sagittal plane on the dorsal and ventral midlines.
Type III	Located at high density loci of superficial nerves and nerve plexuses
Type IV	Located at musculotendinous junctions where the Golgi tendon organs are located.

**Table 2 vetsci-09-00252-t002:** Characteristics of Nerve Fibers (Data adapted from Ref. [30]).

Type of Nerve Fiber	Signal Carried	Myelin Sheath	Diameter (Micrometers)	Conduction Speed (m/s)
A-Beta	Touch	Yes	6–12	35–90
A-Delta	Pain—Mechanical and Thermal	Yes	1–5	5–40
C	Pain-Mechanical, thermal, chemical	No	0.2–1.5	0.5–2

## Data Availability

Not applicable.

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
