# Peer review of "Evidence-Based Application of Acupuncture for Pain Management in Companion Animal Medicine"

_vetsci, 2022, doi:10.3390/vetsci9060252_

Round 1

Reviewer 1 Report

Review

Veterinary Sciences

Evidence-based Application of acupuncture for pain management in companion animal medicine

COMMENTS for Authors:

Thank you for your revisions.  See below for some further comments.

Lines 37-39 – please provide a reference for this statement.

Line 75 – there is an extra “space” after the word organ – please delete.

Figure 1 – the authors in the preceding paragraph mention Extra Meridian points, but yet I do not see that any of these Extra Meridians are labeled on the figure.

Table 2 – fix the formatting between the word conduction and speed.

Overall nice job – much improved!

Author Response

Lines 37-39 – please provide a reference for this statement DONE

Line 75 – there is an extra “space” after the word organ – please delete.DONE

Figure 1 – the authors in the preceding paragraph mention Extra Meridian points, but yet I do not see that any of these Extra Meridians are labeled on the figure.

FIGURE 2 ADDED WITH EXTRA MERIDIAN POINTS

Table 2 – fix the formatting between the word conduction and speed. DONE

Reviewer 2 Report

I am reluctant to give some specific feedback on this paper as it seems to me that the authors performed a literature review about the status quo on using acupuncture in animal medicine.

Could the authors clarify more specifically what is the purpose/objective/aim of the paper? Which are the outcomes?

Author Response

Could the authors clarify more specifically what is the purpose/objective/aim of the paper? Which are the outcomes?

This paper was meant to outline the current status of acupuncture, its scientific evidence and its use for pain control in small animal medicine. The AAHA's previous pain guidelines recommended acupuncture for small animal pain. The current guidelines say there is no evidence. This paper is a reaction to that statement and is meant for general practitioners to realize that acupuncture is an effective form of pain control. This is not meant to be a research project but simply a review.

Round 2

Reviewer 2 Report

Authors properly addressed my main concern, and I am satisfied that this is not mean to be a research-based paper but only a review of the topic’s status quo.

This manuscript is a resubmission of an earlier submission. The following is a list of the peer review reports and author responses from that submission.

Round 1

Reviewer 1 Report

Review

Veterinary Sciences

Evidence-based Application of acupuncture for pain management in companion animal medicine

Abstract:

  • Last line should say – ‘mechanisms of action.’

Introduction/Body:

  • The first 3 paragraphs should be reorganized. The author first defines the word pain, then discusses that acupuncture is used to treat pain, then defines what acupuncture is.  It would make more sense to define acupuncture then discuss its use for pain control.
  • Line 16 – should say ‘drug-free.’
  • Lines 34 and 37 say basically the same thing – please reword.
  • Lines 56-58 – this sentence is confusing, please reword.
  • Line 61 – what do you mean by “extra meridian points?” Please explain.
  • Line 89 – I think it is meant to say: “…act as a form…”
  • Line 113 – This should say: “…in humans and laboratory animals…”
  • Lines 135-140 – the author states that A-Delta fibers are involved in the gate theory of pain control but then does not discuss their involvement, but rather just discusses the involvement of A Beta and C fibers. Please improve this section – it is not clear from the current way this section is written why the characteristics of A Beta fibers vs. C fibers are related to “closing the gate” nor the role of A Delta fibers.  Please make this more clear.
  • Lines 142-144 – it is unclear what the author trying to say in this sentence? Are they trying to say that NMDA receptor activity is decreased by acupuncture because acupuncture increases the levels of serotonin, noradrenaline, and endogenous opioids?  If so, this should be made more clear.
  • Lines 149 – what is meant by the word “Dewey” in parentheses in this sentence?
  • Lines 154-164 – this entire section is unclear. Please reword so that it makes more sense.  I think, at minimum, a definition of ‘segmental dysfunction’ is necessary.
  • Lines 166-227 – this section confuses me. It seems to come out of nowhere.  The author is writing about the mechanisms of action of acupuncture and then it almost seems as though a section heading is missing prior to this block of paragraphs.  This block of paragraphs is written synopses of various publications about the use of acupuncture to treat various orthopedic diseases.  It does not fit, as written, in this MOA section.  Immediately following line 227 the author then picks up discussing MOA of acupuncture again.   Did the author mean to make this be its own section?  Or was the intent to put these paragraphs in this mechanism of action section?  If the latter is the intent they should be deleted or put in a separate section.  Also, I do not think synopses of each of these articles needs to be in the paper, just statements about how acupuncture can successfully be used to treat hip OA or CCL disease and citations but not synopsis paragraphs.
    • In reading through the entire manuscript and going back and reading the abstract and purpose of the paper, as written by the author, I get the impression this section should go at the end after all the MOA of acupuncture have been discussed. Examples of the use of acupuncture could be discussed after the reader understands how acupuncture works. This would make more sense as the purpose of the paper is:  to “outline the mechanism of actions of acupuncture, related neurophysiology and give examples of current literature and studies demonstrating its effective-ness.”
    • Also in this section of paragraphs gold bead therapy and EA is referenced, but no information is given to introduce these topics. These two topics should be introduced in the beginning section in which what acupuncture is is discussed.  If I did not know much about acupuncture and picked up this article to read about its MOA, etc. I would have no idea what gold bead therapy or EA actually were.  Please introduce these topics prior to referencing their use clinically.
  • Line 248 – should say: “…brain segments involved…”
  • Line 250 – to what “suppression” is the author referring? Pain suppression?  Please make this more clear.
  • Line 252 – Should read: “…both of which had analgesic effects.”
  • Line 256 – the author uses the acronym fMRI but I do not think this has been defined for the reader.
  • Line 264-266 – incomplete sentence as written.
  • Lines 250-277 – In this section the suprasegmental effects of acupuncture are discussed but examples used are those in which DN is compared to EA. I would revise this section and find citations that support the heading of this section; citations that prove that one of the MOA of acupuncture is thru stimulation of the brain itself.
  • Lines 279-284 – another contraindication that should be listed is the use of acupuncture in patients with bleeding disorders.
  • Table 2 – I do not think this table adds to the manuscript nor the purpose of the manuscript as discussed in the outline. I would delete this table.
  •  

Overall Impression:

I like the idea of this paper but am very confused by the title, the abstract and the actually body of the manuscript.  The title does not seem to effectively match what the abstract describes the paper to be about.  The paper then goes on to discuss MOA and a few papers on the use of acupuncture for pain control but does not really delve into the wealth of papers on its use for pain control (as the title would imply the paper was going to do).  I would take a global look at the paper and determine what the true goal/purpose of the paper is and revise the title, abstract and body to reflect that goal so that it is more cohesive and easier to read.  Also look at the organization of the body as it does seem like the examples of the effectiveness of acupuncture are randomly put in the middle of the MOA section.  Also, if this is meant to be an introduction to acupuncture, which it seems like it is from the introduction, then things like EA and gold bead therapy need to be discussed and not just randomly mentioned in the body.

Author Response

Abstract:

  • Last line should say – ‘mechanisms of action.’ Corrected

Introduction/Body:

  • The first 3 paragraphs should be reorganized. The author first defines the word pain, then discusses that acupuncture is used to treat pain, then defines what acupuncture is.  It would make more sense to define acupuncture then discuss its use for pain control. I did this as requested however some of the lines you site are now not the same.
  • Line 16 – should say ‘drug-free.’corrected
  • Lines 34 and 37 say basically the same thing – please reword. Done
  • Lines 56-58 – this sentence is confusing, please reword. Done
  • Line 61 – what do you mean by “extra meridian points?” Please explain. This has been reworded
  • Line 89 – I think it is meant to say: “…act as a form…” Corrected
  • Line 113 – This should say: “…in humans and laboratory animals…” Corrected
  • Lines 135-140 – the author states that A-Delta fibers are involved in the gate theory of pain control but then does not discuss their involvement, but rather just discusses the involvement of A Beta and C fibers. Please improve this section – it is not clear from the current way this section is written why the characteristics of A Beta fibers vs. C fibers are related to “closing the gate” nor the role of A Delta fibers.  Please make this more clear.  Corrected
  • Lines 142-144 – it is unclear what the author trying to say in this sentence? Are they trying to say that NMDA receptor activity is decreased by acupuncture because acupuncture increases the levels of serotonin, noradrenaline, and endogenous opioids?  If so, this should be made more clear. Done
  • Lines 149 – what is meant by the word “Dewey” in parentheses in this sentence? Sorry that was the reference ! It has been correctly numbered now.
  • Lines 154-164 – this entire section is unclear. Please reword so that it makes more sense.  I think, at minimum, a definition of ‘segmental dysfunction’ is necessary.Done

Lines 166-227 – this section confuses me. It seems to come out of nowhere.  The author is writing about the mechanisms of action of acupuncture and then it almost seems as though a section heading is missing prior to this block of paragraphs.  This block of paragraphs is written synopses of various publications about the use of acupuncture to treat various orthopedic diseases.  It does not fit, as written, in this MOA section.  Immediately following line 227 the author then picks up discussing MOA of acupuncture again.   Did the author mean to make this be its own section?  Or was the intent to put these paragraphs in this mechanism of action section?  If the latter is the intent they should be deleted or put in a separate section.  Also, I do not think synopses of each of these articles needs to be in the paper, just statements about how acupuncture can successfully be used to treat hip OA or CCL disease and citations but not synopsis paragraphs. Moved this section but I like the little synopsis as often veterinary readers want to know a little more about the studies without reading the papers so I left them in.

    • In reading through the entire manuscript and going back and reading the abstract and purpose of the paper, as written by the author, I get the impression this section should go at the end after all the MOA of acupuncture have been discussed. Examples of the use of acupuncture could be discussed after the reader understands how acupuncture works. This would make more sense as the purpose of the paper is:  to “outline the mechanism of actions of acupuncture, related neurophysiology and give examples of current literature and studies demonstrating its effective-ness.” Moved this section
    • Also in this section of paragraphs gold bead therapy and EA is referenced, but no information is given to introduce these topics. These two topics should be introduced in the beginning section in which what acupuncture is is discussed.  If I did not know much about acupuncture and picked up this article to read about its MOA, etc. I would have no idea what gold bead therapy or EA actually were.  Please introduce these topics prior to referencing their use clinically.Done
  • Line 248 – should say: “…brain segments involved…” done
  • Line 250 – to what “suppression” is the author referring? Pain suppression?  Please make this more clear. done
  • Line 252 – Should read: “…both of which had analgesic effects.” corrected
  • Line 256 – the author uses the acronym fMRI but I do not think this has been defined for the reader. Corrected
  • Line 264-266 – incomplete sentence as written. Corrected
  • Lines 250-277 – In this section the suprasegmental effects of acupuncture are discussed but examples used are those in which DN is compared to EA. I would revise this section and find citations that support the heading of this section; citations that prove that one of the MOA of acupuncture is thru stimulation of the brain itself.Section was revised
  • Lines 279-284 – another contraindication that should be listed is the use of acupuncture in patients with bleeding disorders. done
  • Table 2 – I do not think this table adds to the manuscript nor the purpose of the manuscript as discussed in the outline. I would delete this table. I think this should remain as a supplementary table
  •  

Reviewer 2 Report

Dear Authors,

thank you for submitting this nice review. I liked it a lot I just need to ask to add at line 52 what you stand for TCVM (I think eerybody can understand but it is correct to write extensively in the first presentation of the word).

BW

Author Response

The Terms Traditional Chinese Medicine and Traditional Chinese Veterinary Medicine first appeared in line 24 and 25 so I have added the brackets and short forms there. Thank you for pointing that out.

Reviewer 3 Report

Line 13, I like the concept here but the add on of acupuncture at the end makes the sentence awkward, please reword

Line 26 talking about “westerners” this seems inaccurate and perhaps a bit derogatory, perhaps something similar can be said without labeling people and making it awkward, plus many vets I know fully embrace the concept of acupuncture now so this seems to be an outdated sentence

Line 27 same, I think this sentence can be said better

Line 34 Acupuncture in general involves needles so this isn’t exactly “modern” unless you are going back to ancient times

Line 37 starting the sentence with They makes the sentence awkward, please reword

Line 62 isn’t the point named Shen Shu not Shu Shen?

Line 96 de qi is lower case here and capitalized later, make consistent

Line 113 Studies in both humans and laboratory animals…

Line 185 were commonly used not are

Line 196 German review what?

Line 198 this sentence is worded awkwardly, please reword

Line 237 gastrointestinal no space

Table 2, neuropathic pain PC not PV? Also some have a – and some don’t. Make consistent

Line 278 This paragraph perhaps also mention seizures as a contraindication for head points

Line 288-289 please reword this sentence it is awkward also be aware of tense, should be past tense

The last two sentences don’t really flow well together, please reword
